# PARS: Pseudo-Label Aware Robust Sample Selection for Learning with Noisy Labels

## Abstract

Acquiring accurate labels on large-scale datasets is both time consuming and expensive. To reduce the dependency of deep learning models on learning from clean labeled data, several recent research efforts are focused on learning with noisy labels. These methods typically fall into three design categories to learn a noise robust model: *sample selection* approaches, *noise robust loss* functions, or *label correction* methods. In this paper, we propose **PARS: P**seudo-Label **A**ware **R**obust **S**ample Selection, a hybrid approach that combines the best from all three worlds in a joint-training framework to achieve robustness to noisy labels. Specifically, PARS exploits all training samples using both the raw/noisy labels and estimated/refurbished pseudo-labels via self-training, divides samples into an ambiguous and a noisy subset via loss analysis, and designs label-dependent noise-aware loss functions for both sets of filtered labels. Results show that PARS significantly outperforms the state of the art on extensive studies on the noisy CIFAR-10 and CIFAR-100 datasets, particularly on challenging high-noise and low-resource settings. In particular, PARS achieved an absolute 12% improvement in test accuracy on the CIFAR-100 dataset with 90% symmetric label noise, and an absolute 27% improvement in test accuracy when only 1/5 of the noisy labels are available during training as an additional restriction. On a real-world noisy dataset, Clothing1M, PARS achieves competitive results to the state of the art.

## 1 Introduction

Deep neural networks rely on large-scale training data with human annotated labels for achieving good performance (Deng et al., 2009; Everingham et al., 2010). Collecting millions or billions of labeled training data instances is very expensive, requires significant human time and effort, and can also compromise user privacy (Zheng et al., 2020; Bonawitz et al., 2017). Hence, there has been a paradigm shift in the interests of the research community from large-scale supervised learning (Krizhevsky et al., 2017; He et al., 2016a; Huang et al., 2017) to Learning with Noisy Labels (LNL) (Natarajan et al., 2013; Goldberger & Ben-Reuven, 2016; Patrini et al., 2017; Tanno et al., 2019) and/or unlabeled data (Berthelot et al., 2019b;a; Sohn et al., 2020). This is largely due to the abundance of raw unlabeled data with weak user tags (Plummer et al., 2015; Xiao et al., 2015) or caption descriptions (Lin et al., 2014). However, it is not trivial to build models that are robust to these noisy labels as the deep convolutional neural networks (CNNs) trained with cross-entropy loss can quickly overfit to the noise in the dataset, harming generalization (Zhang et al., 2016).

Most of the existing approaches on LNL can be divided into three main categories. First, several *noise robust loss* functions (Ghosh et al., 2017; Wang et al., 2019a; Zhang & Sabuncu, 2018) were proposed that are inherently tolerant to label noise. Second, *sample selection* methods (also referred to as *loss correction* in some literature) (Han et al., 2018; Yu et al., 2019; Arazo et al., 2019) are a popular technique that analyzes the per-sample loss distribution and separates the clean and noisy samples. The identified noisy samples are then re-weighted so that they contribute less in the loss computation. A challenge in this direction is to design a reliable criterion for separation and hence prevent overfitting to highly confident noisy samples, a behavior known as *self-confirmation bias*. Third, *label correction* methods attempt to correct the noisy labels using class-prototypes (Han et al., 2019) or pseudo-labeling techniques (Tanaka et al., 2018; Yi & Wu, 2019). However, in order to correct noisy labels, we typically need an extra (usually small) set of correctly labeled validation

labels. In particular, these methods can fail when the noise ratio is high and estimating correct labels or high-quality pseudo-labels is non-trivial.

More recently, the success of several state-of-the-art LNL methods is attributed to leveraging Semi-Supervised Learning (SSL) based approaches (Li et al., 2020; Kim et al., 2019). Typically, a sample selection technique is applied to separate clean and noisy labels in the training data, then the noisy labels are deemed unreliable and hence treated as unlabeled in a SSL setting. Following the recent SSL literature (Lee et al., 2013; Arazo et al., 2020), estimated pseudo-labels are usually used to replace the filtered noisy labels during training. These approaches have shown to be highly tolerant to label-noise. However, the noisy labels are always discarded in favor of pseudo-labels in all the existing literature, but they may still contain useful information for training. Pseudo-labeling is in turn only applied to the filtered noisy subset while the rest of the raw labels are typically used as is, which makes it sensitive to the quality of the filtering algorithm.

In particular, motivated by a simple principle of making the most of the signal contained in the noisy training data, we design PARS, short for **P**seudo-Label **A**ware **R**obust **S**ample Selection. Our contributions are as follows:

1. PARS proposes a novel, principled training framework for LNL. It trains on both the original labels and pseudo-labels. Unlike previous works, instead of filtering and then discarding the low-confident noisy labels, PARS uses the entire set of original labels, and applies self-training with pseudo-labeling and data augmentation for the entire dataset (rather than the filtered noisy data only).

2. PARS is able to learn useful information from all the available data samples through label-dependent noise-aware loss functions. Specifically, in order to prevent overfitting to inaccurate original labels (or inaccurate pseudo-labels), PARS performs a simple confidence-based filtering technique by setting a high threshold on their predicted confidence, and applies robust/negative learning (or positive/negative learning) accordingly.

3. We perform extensive experiments on multiple benchmark datasets *i.e.* noisy CIFAR-10, noisy CIFAR-100 and Clothing1M. Results demonstrate that PARS outperforms previous state-of-the-art methods by a significant margin, in particular when high level of noise is present in the training data. We also conduct sufficient ablation studies to validate the importance of our contributions.

4. We design a novel *low-resource semi-supervised LNL* setting where only a small subset of data is weakly labeled (Section 4.3). We show significant gains over state-of-the-art approaches using PARS. This setting is particularly interesting when it is hard to obtain large-scale noisy labeled data. In particular, we find that surprisingly *none* of the existing LNL methods outperform a baseline SSL model (FixMatch) (Sohn et al., 2020) that is not even designed to handle label noise, and yet PARS can achieve up to an absolute 27% improvement in test accuracy in a controlled high-noise low-resource setting.

## 2 RELATED WORK

In recent literature on LNL, methods typically fall into three design categories to learn a noise robust model: *noise robust loss* functions, *sample selection* approaches, or *label correction* methods.

*Noise robust loss* function based methods propose objective functions that are tolerant to label noise. A commonly used loss function that is identified to be robust to noisy labels is Mean Absolute Error (MAE) (Ghosh et al., 2017). Wang et al. (2019a) proposed Improved MAE which is a re-weighted version of MAE. Zhang & Sabuncu (2018) proposed Generalized Cross Entropy Loss (GCE) which is a generalization of MAE and Categorical Cross Entropy loss. More recently, Wang et al. (2019b) designed a Symmetric Cross Entropy (SCE) loss which is similar in spirit to the symmetric KL-divergence and combines the cross-entropy loss with the reverse cross-entropy. Although SCE is robust to noisy labels, Ma et al. (2020) proposed a normalized family of loss functions which are shown to be more robust than SCE for extreme levels of label noise. Kim et al. (2019) and Kim et al. (2021) designed a framework that alternates between positive learning on accurate/clean labels and negative learning on complementary/wrong labels. *Loss correction* approaches explicitly modify the loss function during training to take into account the noise distribution by modeling a noise transition

matrix (Patrini et al., 2017; Tanno et al., 2019; Xia et al., 2019; Goldberger & Ben-Reuven, 2016) or based on label-dependent weights (Natarajan et al., 2013).

Another family of methods focus primarily on *sample selection*, where the model selects small-loss samples as "clean" samples under the assumption that the model first fits to the clean samples before memorizing the noisy samples (also known as the early-learning assumption) (Arpit et al., 2017; Zhang et al., 2016; Liu et al., 2020). Han et al. (2018) and Yu et al. (2019) proposed Co-teaching where sample selection is conducted using two networks to select clean and noisy samples, and then the clean samples are used for further training. MentorNet (Jiang et al., 2018) is a student-teacher framework where a pre-trained teacher network guides the learning of the student network with clean samples (whose labels are deemed "correct"). In the decoupling training strategy proposed by Malach & Shalev-Shwartz (2017), two networks are trained simultaneously and guide each other on when and how to update. One limitation of these approaches is that they ignore all the noisy/unclean samples during training and only leverage the expected-to-be clean samples for improving performance. Li et al. (2020) proposed DivideMix where sample selection is conducted based on per-sample loss distribution, and then the noisy samples are treated as unlabeled data in a SSL setting (Berthelot et al., 2019b). Nishi et al. (2021) further investigated the potential of using augmentation strategies in LNL, one used for loss analysis and another for learning, thus improving generalization of DivideMix.

Compared to the above, *label correction* aims to improve the quality of the noisy labels by explicitly correcting the wrong labels. Tanaka et al. (2018) and Yi & Wu (2019) predicted the correct labels either as estimates of label probabilities (soft labels) or as one-hot class labels (hard labels). Arazo et al. (2019) combined label correction with iterative sample selection by first filtering clean labels from noisy labels modeling a two-component Beta Mixture, and then estimating to correct labels for those noisy samples. Tanaka et al. (2018) combined label correction with additional regularization terms that were proved to be helpful for LNL. Song et al. (2019) also used label replacement to refurbish a subset of labels, thereby gradually increasing the number of available training samples. Liu et al. (2020) computed soft labels as model predictions and then exploit it to avoid memorization.

## 3 METHODOLOGY

### 3.1 PRELIMINARIES

For a $K$-class classification problem with noisy labels, let $\mathcal{D} = \left\{(\boldsymbol{x}^i, y^i)\right\}_{i=1}^n$ with $\boldsymbol{x} \in \mathcal{X} \subset \mathbb{R}^d$ as the input features and $y \in \mathcal{Y} = \{1, \ldots, K\}$ as the corresponding label, we aim to learn a classifier $p(\cdot; \theta) : \mathcal{X} \to \mathcal{Y}$ parametrized by $\theta$. For a clean labeled set, *i.e.* where $y^i$ is the true label for the $i$-th training sample $\boldsymbol{x}^i$, we can learn the model parameters $\theta$ by minimizing the Cross Entropy (CE) loss:

$$\min_{\theta} \frac{1}{|\mathcal{D}|} \sum_{(\boldsymbol{x}, y) \in \mathcal{D}} \mathcal{L}_{CE}(p(\boldsymbol{x}; \theta), y) . \tag{1}$$

However, in the presence of noisy labels where $y^i$ is possibly incorrect for the $i$-th training sample $\boldsymbol{x}^i$, the cross-entropy loss can quickly overfit to the noise in the dataset (Zhang et al., 2016).

**Robust Learning.** Variants to the CE loss have been proposed to improve the classification performance under label noise. Examples of such noise robust loss functions include Mean Absolute Error $\mathcal{L}_{MAE}$ (Ghosh et al., 2017), Symmetric Cross Entropy $\mathcal{L}_{SCE}$ (Wang et al., 2019b), Normalized Cross Entropy $\mathcal{L}_{NCE}$ or Active Passive Loss $\mathcal{L}_{APL}$ (Ma et al., 2020). We may simply replace the $\mathcal{L}_{CE}$ with any of these noise robust loss, denoted by $\mathcal{L}_{RL}$, in the optimization Equation (1) to perform Robust Learning:

$$\mathcal{L}_{RL} \in \{\mathcal{L}_{MAE}, \mathcal{L}_{SCE}, \mathcal{L}_{NCE}, \mathcal{L}_{APL}\} . \tag{2}$$

For the sake of the limited space, definitions and further discussion about the robust losses are deferred to Appendix A.1.

**Positive and Negative Learning**. A particular variant to the CE loss that is designed to handle noisy/wrong labels is called Negative Learning (Kim et al., 2019). Given a sample with a correct label $(\boldsymbol{x}, y)$, *Positive Learning* (PL) is equivalent to optimizing the standard CE loss in Equation (1), so that we optimize the output probabilities corresponding to the true label to be close to 1:

$$\mathcal{L}_{PL}(p(\boldsymbol{x}; \theta), y) := \mathcal{L}_{CE}(p(\boldsymbol{x}; \theta), y) = \sum_{k=1}^{K} [y]_k \log [p(\boldsymbol{x}; \theta)]_k , \tag{3}$$

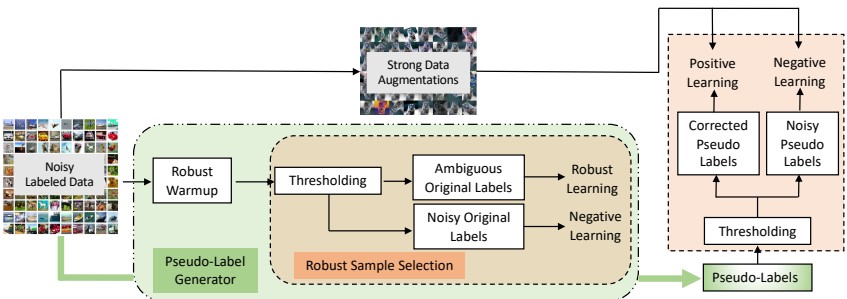

Figure 1: PARS: **P**seudo-Label **A**ware **R**obust **S**ample Selection. PARS has two stages: (1) First we perform robust warm-up training. (2) We then iteratively obtain pseudo-labels on-the-fly and perform a joint training with the original raw labels and self-training with pseudo-labels, filtering the raw labels into ambiguous/noisy samples and the pseudo-labels into corrected/noisy samples. We apply label-dependent noise-aware loss functions in a robust/negative and positive/negative learning framework for the raw labels and pseudo-labels respectively.

where the class label $y$ is one-hot encoded, and $[\cdot]_k$ denotes the $k$-th dimension of a vector. Now for a data example with a wrong label $(\boldsymbol{x}, \bar{y})$, *Negative Learning* (NL) applies the CE loss function to fit the "complement" of the label in Equation (1), so that the predicted probabilities corresponding to the given label is optimized to be far from 1:

$$\mathcal{L}_{NL}(p(\boldsymbol{x}; \theta), \bar{y}) := -\sum_{k=1}^{K} [\bar{y}]_k \log \left[1 - p(\boldsymbol{x}; \theta)\right]_k . \tag{4}$$

**Sample Selection.** In order to apply the Positive and Negative Learning framework in LNL, Kim et al. (2019) proposed to select samples based on a simple threshold of the per-sample loss value, *i.e.* a data point $(\boldsymbol{x}, y)$ is selected to be clean for PL if $[p(\boldsymbol{x}; \theta)]_y > \gamma$ where $\gamma$ is a noise-aware threshold. This simple yet effective procedure is based on the commonly used "small-loss" assumption that smaller-loss samples (typically in early learning stages) are usually easier to learn and hence expected to be "clean", and it has been successfully applied in several other studies (Jiang et al., 2018; 2020).

## 3.2 PARS: Pseudo-Label Aware Robust Sample Selection

We propose PARS: **P**seudo-Label **A**ware **R**obust **S**ample Selection. The method is illustrated in Figure 1 (and Algorithm 1 in Appendix A.2), and each component is discussed in detail below. In particular, there are three major differences between PARS and previous works: 1) PARS does not treat the high-confidence samples as "clean", but rather applies robust loss functions to address the relatively low yet non-negligible level of noise. 2) PARS does not discard the noisy (or low-confidence) labels, but rather applies a negative learning framework to learn the "complement" of these labels (Kim et al., 2019). This constitutes one major difference between the selective negative learning procedure of Kim et al. (2019) and PARS, that Kim et al. (2019) only apply it during the warm-up stage and still choose to discard the filtered noisy labels in favour of pseudo-labels in a SSL manner, and we unify the two stages and show improved performance. 3) We improve the convergence of PARS by self-training using pseudo-labels and data augmentation for the entire dataset as a proxy to enhance clean labels and correct noisy labels.

**Robust Warm-up Training.** In the presence of label noise, the model overfits to noise when trained with CE loss and produces overconfident and wrong predictions after a few epochs. This problem is more severe with high levels of label noise. To overcome this, we perform "robust warm-up" using a robust loss for a few epochs for the model to produce reliable and confident predictions. Specifically, in the warm-up training phase, we minimize:

$$\min_{\theta} \frac{1}{|\mathcal{D}|} \sum_{(\boldsymbol{x}, y) \in \mathcal{D}} \mathcal{L}_{RL}(p(\boldsymbol{x}; \theta), y), \tag{5}$$

where $\mathcal{L}_{RL}$ can be any loss from Equation (2). In our experiments, we test different loss functions for warm-up training (for more details see Appendix A.3).

**Robust Sample Selection.** As shown in Figure 2a, after the robust warm-up training, we observe that there is a weak trend that high-confidence samples tend to have clean labels. Given the model

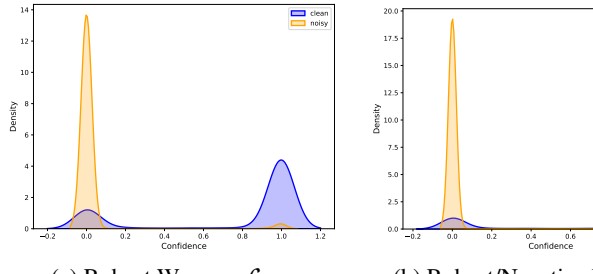
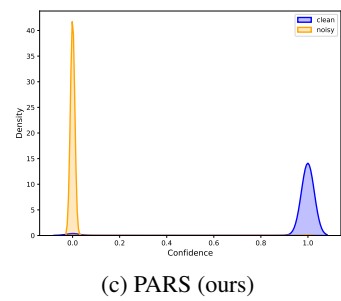

(a) Robust Warmup $\mathcal{L}_{RL}$     (b) Robust/Negative Learning     (c) PARS (ours)

Figure 2: Confidence Probabilities for different stages of training on CIFAR-10 dataset with 50% symmetric noise. PARS (ours) improves separation of the clean and noisy labels as compared to the model after the robust warm-up training.

parameters $\hat{\theta}$ at the current iteration and a confidence threshold $\tau \in [0, 1]$, we divide the entire dataset $\mathcal{D}$ (or similarly, a minibatch $\mathcal{B} \subset \mathcal{D}$) into an *ambiguous*[1] set $\mathcal{D}_A$ and *noisy* set $\mathcal{D}_N$ of samples by thresholding the *maximum* of the predicted confidence probabilities over all classes:

$$\mathcal{D}_A = \left\{ (\boldsymbol{x}, y) \in \mathcal{D} \,\middle|\, \max_k \left[ p(\boldsymbol{x}; \hat{\theta}) \right]_k > \tau \right\}, \quad \mathcal{D}_N = \mathcal{D} \setminus \mathcal{D}_A \,. \tag{6}$$

A notable difference is that our confidence-based thresholding is label-free (*i.e.*thresholding of the maximum confidence over all classes), while the widely used loss-based thresholding Kim et al. (2019) depends on the ground-truth (noisy) label (*i.e.*thresholding the confidence corresponding to the given class). Our approach has two advantages: 1) our selective criterion depends on the predicted confidence probabilities alone, therefore it is straightforward to apply to cases where ground-truth labels are unavailable (*e.g.*in a semi-supervised setting in Section 4.3); 2) our strategy helps to reduce the self-confirmation bias of propagating the mistake of overfitting to possibly noisy samples, because our model only selects samples that it is confident to make a prediction on regardless of the correctness of the raw label.

Notably, previous works such as Jiang et al. (2018); Li et al. (2020) usually maintain two divergent networks (one for sample selection and one for the main classification task) in order to avoid the self-confirmation bias. Our method proves to be highly resilient to the overfitting issue with a single network simultaneously updated to perform both sample selection and classification, which drastically simplifies implementation and training complexity.

**Label-Dependent Noise-Aware Loss Functions.** Despite the robust sample selection of dividing samples into ambiguous and noisy set, we propose to learn useful information from all the available training data instead of discarding the raw labels of the noisy samples as typically done in previous literature (Jiang et al., 2018; Kim et al., 2019; Li et al., 2020).

Specifically, we propose to use two separate loss functions, tailored for either the ambiguous or noisy labels respectively, which prevents overfitting to noise and enables further filtering of labels effectively (Figure 2b). For the ambiguous set, $\mathcal{D}_A$, we train with the robust learning loss, $\mathcal{L}_{RL}$, (same loss used during the warm-up training), where we deem that the given label in this set may still contain non-negligible noise. For the noisy set, $\mathcal{D}_N$, we train with the negative learning loss, $\mathcal{L}_{NL}$, to learn the complement of the given raw labels for the identified least-confident highly-noisy samples. The overall label-dependent noise-aware loss function for the proposed robust/negative learning framework using the given raw labels is thus:

$$\mathcal{L}_{raw}(\theta) := \left( \frac{1}{|\mathcal{D}_A|} \sum_{(\boldsymbol{x}, y) \in \mathcal{D}_A} \mathcal{L}_{RL}(p(\boldsymbol{x}; \theta), y) \right) + \lambda_N \left( \frac{1}{|\mathcal{D}_N|} \sum_{(\boldsymbol{x}, \bar{y}) \in \mathcal{D}_N} \mathcal{L}_{NL}(p(\boldsymbol{x}; \theta), \bar{y}) \right), \tag{7}$$

where $\lambda_N$ is the weight for the negative learning loss term, and $\mathcal{D}_A, \mathcal{D}_N$ are defined as in Equation (6).

We set a high threshold (*e.g.*0.95 throughout our experiments) on the predicted confidence in the robust sample selection (Equation (6)) since we observe that this yields better performance as shown

---

[1]We choose the term "ambiguous" rather than the typically used term "clean" (Jiang et al., 2018; 2020) because there is still significant amount of noise in the expected-to-be clean samples when a simple thresholding method is used to select samples (Arazo et al., 2019).

in Appendix A.4. We will show that, by using the noisy labels in the negative learning setting, we are able to consistently improve the model performance (Table 4).

**Self-Training with Pseudo-Labels.** After the robust warm-up training, the initial convergence of the model leads to reliable predictions which can be used as labels to further improve model convergence via self-training. Therefore, in parallel to using the given raw labels during training, we use pseudo-labels generated by the network itself to guide the learning process. Given the model parameters $\hat{\theta}$ at the current iteration, for each sample $(\boldsymbol{x}, \cdot) \in \mathcal{D}$, we generate the underlying pseudo-label $z$ corresponding to the model's most confident class, and then randomly sample a complementary "wrong" pseudo-label $\bar{z}$ from the remaining classes following Kim et al. (2019):

$$z := \arg\max_k \left[ p(\boldsymbol{x}; \hat{\theta}) \right]_k, \quad \bar{z} \in \{1, \dots, K\} \setminus \{z\}. \tag{8}$$

Similarly to how we use label-dependent noise-aware loss functions for the given raw labels in the ambiguous and noisy sets, we train with the pseudo-labels from the two sets using different losses too. Specifically, we apply positive/negative learning for the generated pseudo-labels in the ambiguous/noisy set respectively:

$$\mathcal{L}_{pseudo}(\theta) := \left( \frac{1}{|\mathcal{D}_A|} \sum_{(\boldsymbol{x}, \cdot) \in \mathcal{D}_A} \mathcal{L}_{PL}(p(\boldsymbol{x}; \theta), z) \right) + \lambda_N \left( \frac{1}{|\mathcal{D}_N|} \sum_{(\boldsymbol{x}, \cdot) \in \mathcal{D}_N} \mathcal{L}_{NL}(p(\boldsymbol{x}; \theta), \bar{z}) \right), \tag{9}$$

where $\lambda_N$ is the weight for the negative learning loss term (equal to the weight in Equation (7)), $\mathcal{D}_A, \mathcal{D}_N$ are defined as in Equation (6), $z, \bar{z}$ are defined as in Equation (8). We find that it suffices to train with the standard CE loss for self-training with pseudo-labels from the ambiguous set, without the need for noise robust learning.

One caveat of applying self-training is that, the model's own predictions may become unreliable when most of the samples are primarily guided by the self-training, encouraging the network to predict the same class to minimize the loss. We observe that this is particularly the case under high levels of label noise. To reliably apply self-training, we regularize Equation (9) with a confidence penalty (Tanaka et al., 2018; Arazo et al., 2019):

$$\mathcal{L}_{reg}(\theta) := \sum_{k=1}^{K} [p]_k \log \left( \frac{[p]_k}{[h(\mathcal{D}; \theta)]_k} \right), \tag{10}$$

where $[p]_k$ is the prior probability distribution for class $k$, and $[h(\mathcal{D}; \theta)]_k$ is shorthand for the mean predicted probability of the model for class $k$ across all samples in the dataset $\mathcal{D}$. In our experiments, we assume a uniform distribution for the prior probabilities (*i.e.* $[p]_k = 1/K$), while approximating $h(\mathcal{D}; \theta)$ by $h(\mathcal{B}; \theta)$ using mini-batches $\mathcal{B} \subset \mathcal{D}$ as suggested by Tanaka et al. (2018). The generated pseudo-labels become reliably confident with help of the regularization, which guides the robust sample selection to resemble the true distribution more closely (Figure 2c).

**PARS: Pseudo-Label Aware Robust Sample Selection.** After the robust warm-up training stage (Equation (5)), our final model, PARS, is then trained using both the original labels and the self-generated pseudo-labels with robust sample selection. The final loss is given by:

$$\min_\theta \mathcal{L}_{total}(\theta) := \mathcal{L}_{raw}(\theta) + \lambda_S \mathcal{L}_{pseudo}(\theta) + \lambda_R \mathcal{L}_{reg}(\theta), \tag{11}$$

where $\mathcal{L}_{raw}, \mathcal{L}_{pseudo}, \mathcal{L}_{reg}$ are defined as in Equations (7), (9) and (10) respectively, and $\lambda_S, \lambda_R$ are the weights for the self-training loss and confidence penalty term.

**Data Augmentation.** For image classification tasks in LNL, *data augmentation* techniques have recently been shown to significantly improve generalization (Nishi et al., 2021). We adapt such data augmentation strategies to further improve PARS for benchmarking image classification performance in our experiments. Specifically, we apply *weak augmentation* (*e.g.* standard random flip, crop, normalization) to all data samples in the robust/negative learning with original raw labels in computing Equation (7). We also apply *strong augmentation* (*e.g.* AutoAugment (Cubuk et al., 2019), RandAugment (Cubuk et al., 2020)) to all data samples in the positive/negative learning with pseudo-labels in computing Equations (9) and (10). We do not incorporate strong augmentation for training with the original labels as it leads to poor performance in presence of high label noise, as also observed by Nishi et al. (2021). In this way, PARS can be seen as one example of how to extend the work of Nishi et al. (2021) by combining weak and strong data augmentation effectively with noisy labels and pseudo-labeling, as an effective technique to further advance the state of the art in LNL.

| Datasets | CIFAR-10 | | | | | CIFAR-100 | | | |
|---|---|---|---|---|---|---|---|---|---|
| Noise Type | Sym. | | | | Asym. | Sym. | | | |
| Alg.\Noise Ratio | 20% | 50% | 80% | 90% | 40% | 20% | 50% | 80% | 90% |
| Cross-Entropy Loss $\mathcal{L}_{CE}$ | 86.8 | 79.4 | 62.9 | 42.7 | 85.0 | 62.0 | 46.7 | 19.9 | 10.1 |
| Bootstrap (Reed et al., 2014) | 86.8 | 79.8 | 63.3 | 42.9 | - | 62.1 | 46.6 | 19.9 | 10.2 |
| F-correction (Patrini et al., 2017) | 86.8 | 79.8 | 63.3 | 42.9 | 87.2 | 61.5 | 46.6 | 19.9 | 10.2 |
| Mixup (Zhang et al., 2017) | 95.6 | 87.1 | 71.6 | 52.2 | - | 67.8 | 57.3 | 30.8 | 14.6 |
| Co-teaching+ (Yu et al., 2019) | 89.5 | 85.7 | 67.4 | 47.9 | - | 65.6 | 51.8 | 27.9 | 13.7 |
| P-correction (Yi & Wu, 2019) | 92.4 | 89.1 | 77.5 | 58.9 | 88.5 | 69.4 | 57.5 | 31.1 | 15.3 |
| Meta-Learning (Li et al., 2019) | 92.9 | 89.3 | 77.4 | 58.7 | 89.2 | 68.5 | 59.2 | 42.4 | 19.5 |
| M-correction (Arazo et al., 2019) | 94.0 | 92.0 | 86.8 | 69.1 | 87.4 | 73.9 | 66.1 | 48.2 | 24.3 |
| DivideMix (Li et al., 2020) | 96.1 | 94.6 | 93.2 | 76.0 | 93.4 | 77.3 | 74.6 | 60.2 | 31.5 |
| ELR+ (Liu et al., 2020) | 95.8 | 94.8 | 93.3 | 78.7 | 93.0 | 77.6 | 73.6 | 60.8 | 33.4 |
| DM-AugDesc-WS-WAW (Nishi et al., 2021) | **96.3** | 95.4 | 93.8 | 91.9 | 94.6 | 79.5 | 77.2 | 66.4 | 41.2 |
| DM-AugDesc-WS-SAW (Nishi et al., 2021) | **96.3** | 95.6 | 93.7 | 35.3 | 94.6 | **79.6** | **77.6** | 61.8 | 17.3 |
| Robust Loss $\mathcal{L}_{RL}$ (warm-up training only) | 84.0 | 76.4 | 74.1 | 54.5 | 79.0 | 63.9 | 59.8 | 42.1 | 18.9 |
| **PARS** (ours) | **96.3** | **95.8** | **94.9** | **93.6** | **95.4** | 77.7 | 75.3 | **69.2** | **53.1** |

Table 1: Test Accuracy (%) for all the baselines and our proposed method, PARS, on CIFAR-10 and CIFAR-100 with symmetric noise (noise ratio ranging 20% to 90%) and asymmetric noise (noise ratio 40%) on CIFAR-10. The results above the "DivideMix" row are taken from Li et al. (2020).

| Alg.\Dataset | Clothing1M |
|---|---|
| Cross-Entropy Loss $\mathcal{L}_{CE}$ | 69.21 |
| F-correction (Patrini et al., 2017) | 69.84 |
| M-correction (Arazo et al., 2019) | 71.00 |
| Joint-Optim (Tanaka et al., 2018) | 72.16 |
| Meta-Learning (Li et al., 2019) | 73.47 |
| P-correction (Yi & Wu, 2019) | 73.49 |
| DivideMix (Li et al., 2020) | 74.76 |
| ELR+ (Liu et al., 2020) | 74.81 |
| DM-AugDesc-WS-WAW (Nishi et al., 2021) | 74.72 |
| DM-AugDesc-WS-SAW (Nishi et al., 2021) | **75.11** |
| Robust Loss $\mathcal{L}_{RL}$ (warm-up training only) | 71.80 |
| **PARS (ours)** | 74.61 |

Table 2: Test Accuracy (%) for all the baselines and our proposed method, PARS, on Clothing1M. The results above the "DivideMix" row are taken from Li et al. (2020).

# 4 EXPERIMENTS

## 4.1 DATASETS AND IMPLEMENTATION DETAILS

We perform extensive experiments on two benchmark datasets, CIFAR-10 and CIFAR-100 (Krizhevsky et al., 2009) with synthetic label noise. The CIFAR-10 and CIFAR-100 datasets each contain 50,000 training images and 10,000 test images of size $32 \times 32$ pixels with 10 and 100 classes respectively. Both of these datasets have clean label annotations, hence we inject synthetic noise to train the models, and test the performance using the original clean labels. Following previous work (Arazo et al., 2019; Li et al., 2020), we evaluate with two types of noise: symmetric (uniform) and asymmetric (class-conditional). For symmetric noise, the labels are generated by flipping the labels of a certain percentage of training samples to any other class labels uniformly. We use noise ratios of $\{0.2, 0.5, 0.8, 0.9\}$ which are the symmetric noise probabilities as defined in (Patrini et al., 2017; Li et al., 2020). While for asymmetric noise labels, flipping of labels only happens for a certain set of classes (Patrini et al., 2017; Li et al., 2020) such as *truck* $\rightarrow$ *automobile*, *bird* $\rightarrow$ *airplane*, *deer* $\rightarrow$ *horse*, and *cat* $\leftrightarrow$ *dog*. For asymmetric noise, we use the noise rate of 40%.

We also test our method on the Clothing1M dataset (Xiao et al., 2015) which is a large-scale real-world noisy dataset containing 1 million images from online shopping websites. The labels are collected by extracting tags from the surrounding texts and keywords, and can thus be noisy. As the images vary in size, they are resized to $256 \times 256$ pixels. The implementation details for all the experiments are given in Appendix A.3.

## 4.2 COMPARISON WITH THE STATE OF THE ART ON LEARNING WITH NOISY LABELS

In Table 1, we compare our final method PARS with several existing state-of-the-art methods on LNL. For this setting, we evaluate our models following previous work (Liu et al., 2020; Li et al., 2020), where noisy labels are present for a large number of data samples. We evaluate with different

| Dataset | CIFAR-10 | | CIFAR-100 | |
| --- | --- | --- | --- | --- |
| # Labels | 4000 | | 10000 | |
| Noise Type | Sym. | | Sym. | |
| Alg.\Noise Ratio | 50% | 90% | 50% | 90% |
| FixMatch[†] (Sohn et al., 2020) | 82.7 | 54.9 | 55.6 | 22.0 |
| DivideMix[†] (Li et al., 2020) | 66.0 | 37.1 | 25.2 | 8.0 |
| AugDesc-WS[†] (Nishi et al., 2021) | 73.3 | 56.1 | 31.1 | 10.1 |
| DivideMix+[†] (Li et al., 2020) | 74.3 | 40.7 | 10.6 | 7.2 |
| AugDesc-WS+[†] (Nishi et al., 2021) | 73.2 | 57.5 | 30.1 | 7.6 |
| **PARS** (ours) | **95.3** | **70.3** | **76.3** | **48.9** |

Table 3: Test Accuracy (%) for all the baselines and our proposed method, PARS, on CIFAR-10/100 datasets with 50% and 90% symmetric noise with a subset of training labels. $+$ denotes that methods are adapted to include unlabeled data while training. $\dagger$ denotes re-training with the open-source code.

levels of symmetric label noise ranging from 20% - 90% for CIFAR-10/100, and on asymmetric label noise for CIFAR-10 with 40% noise ratio. All of the reported results are the *best* test accuracy of CIFAR-10/100 across all epochs, following the same logic of Li et al. (2020). As shown, compared to all the existing methods, PARS consistently outperforms on CIFAR-10 dataset with different types and levels of label noise. Specifically, we observe an increase over the best reported results by Liu et al. (2020) by approx 14.9% and by approx 1.7% on Nishi et al. (2021) for 90% symmetric label noise, which shows the robustness of our method to high levels of label noise. On a more challenging setting of CIFAR-100 dataset, we improve over the state-of-the-art work of Nishi et al. (2021) by an absolute 12% improvement in test accuracy in the presence of 90% symmetric label noise. Notably, compared to training with a robust loss function alone (*i.e.* performing warm-up training thoroughly), PARS is able to achieve up to 40% absolute improvement in test accuracy. These experimental results strongly advocate for PARS, particularly the proposed training framework of using label-dependent noise-aware loss functions on the ambiguous/noisy original raw labels (and corrected/noisy pseudo-labels) with robust/negative learning (and positive/negative self-learning).

On a noisy large-scale real dataset, Clothing1M, Table 2 shows the best accuracy for PARS and all previous works. In particular, PARS achieve improvements over the the robust training baseline with $\mathcal{L}_{RL}$ by 2.8%. Compared to the state of the art, PARS is able to achieve comparable results with DivideMix (Li et al., 2020) and DM-AugDesc-WS-WAW (Nishi et al., 2021). Another variant method proposed by Nishi et al. (2021), DM-AugDesc-WS-SAW, seemed to obtain the best performance overall due to the use of strong augmentation in a warm-up training phase. It is important to note that both methods from Nishi et al. (2021) are based on DivideMix (Li et al., 2020), whereas PARS is a stand-alone novel training procedure which is able to achieve significant improvements. This paves way to designing other methods for advancing the state of the art in LNL. We will leave to future work the investigation of how to optimize the augmentation strategies in order to further improve the warm-up training of PARS.

### 4.3 Low-resource Semi-supervised Learning with Noisy Labels

We also evaluate our proposed method on a novel *low-resource* LNL in a semi-supervised setting. We randomly take a subset of 4000/10000 samples from the clean labeled CIFAR-10/100 datasets, and then introduce 50% and 90% symmetric noise in these labels, and include all the remaining samples as unlabeled data for training. As none of the previous LNL works evaluate in a low-resource setting, we decide to compare to a state-of-the-art semi-supervised learning method (FixMatch (Sohn et al., 2020)) as baseline, as well as two state-of-the-art methods from LNL (DivideMix (Li et al., 2020), AugDesc-WS (Nishi et al., 2021)). Note that both DivideMix and AugDesc-WS are methodologically motivated by the semi-supervised literature, where the entire training set is filtered into clean and noisy set, and then the noisy samples are simply treated as unlabeled data in a semi-supervised manner. For fair comparison, we also extend these two methods by combining the unlabeled data with the filtered noisy data in their semi-supervised training framework, and term them as "DivideMix+" and "AugDesc-WS+" respectively to signify the difference. All baselines are retrained using the open-sourced code. For PARS, after the warm-up training on the subset of labeled data, we predict pseudo-labels on the unlabeled subset, perform sample selection on the entire dataset (both labeled and unlabeled), and train using optimization objective Equation (11).

As shown in Table 3, the existing LNL models both perform significantly worse than the SSL baseline, FixMatch. This is surprising due to the fact that FixMatch was not designed with consideration of how

| Dataset | CIFAR-10 | | | | | CIFAR-100 | | | |
|---|---|---|---|---|---|---|---|---|---|
| Noise Type | Sym. | | | | Asym. | Sym. | | | |
| Alg.\Noise Ratio | 20% | 50% | 80% | 90% | 40% | 20% | 50% | 80% | 90% |
| Cross-Entropy Loss $\mathcal{L}_{CE}$ | 86.8 | 79.4 | 62.9 | 42.7 | 85.0 | 62.0 | 46.7 | 19.9 | 10.1 |
| Robust Loss $\mathcal{L}_{RL}$ (warm-up training only) | 84.0 | 76.4 | 74.1 | 54.5 | 79.0 | 63.9 | 59.8 | 42.1 | 18.9 |
| PARS w/o $\mathcal{L}_{pseudo}$ | 87.6 | 86.6 | 79.7 | 59.1 | 84.3 | 66.4 | 62.0 | 45.7 | 21.4 |
| PARS w/o $\mathcal{L}_{NL}$ in either $\mathcal{L}_{raw}$ or $\mathcal{L}_{pseudo}$ | 95.6 | 94.6 | 94.3 | 92.4 | 92.3 | 76.6 | 74.3 | 67.0 | 51.3 |
| **PARS (full model)** | **96.3** | **95.8** | **94.9** | **93.6** | **95.4** | **77.7** | **75.3** | **69.2** | **53.1** |

Table 4: Test Accuracy (%) reported on the ablations of PARS on the CIFAR-10/100 datasets.

to handle label noise. Further, both of the existing LNL models perform similarly, sometimes worse, when the unlabeled data are introduced to the training set, meaning that they cannot effectively exploit the signal contained in all available data. However, our method, PARS. significantly outperforms the baselines in all datasets and noise settings by a margin, and achieves an absolute improvement of up to 27% in test accuracy (CIFAR-100 with 90% noise). We deem that this is attributed to the ability of PARS exploiting the pseudo-labels with the positive/negative learning framework, which is robust to a variety of levels and types of label noise in this low-resource LNL setting.

## 4.4 ABLATION STUDY

Table 4 analyzes the results of different components in PARS and how each learning stage contributes to the final model performance. First, we empirically test the performance of the "warm-up training only" stage (without the sample selection or losses on either the original or pseudo labels). This is empirically equivalent to replacing the standard cross-entropy loss $\mathcal{L}_{CE}$ by a noise robust loss $\mathcal{L}_{RL}$ and optimizing Equation (5) thoroughly. With such a noise robust loss function, we achieve good performance compared to using cross-entropy, especially in the presence of high levels of label noise. However, the warmed-up model performs far worse than the "full model" of PARS, emphasizing the importance of robust sample selection and using label-dependent noise-aware losses on both the original and pseudo labels as we have proposed.

Next, we investigate the importance of different components in PARS. 1) Self-training with pseudo-labels is an important part of PARS. Compared to the full model, removing the use of pseudo-labels entirely (*i.e.*"PARS w/o $\mathcal{L}_{pseudo}$") leads to a significant reduction in model performance. This implies that adding the self-training losses to PARS is critical to improve the model convergence after the robust warm-up training. 2) Motivated by the fact that there is exploitable signal contained in the filtered noisy set after robust sample selection, we include information from *all* the original and pseudo labels to enhance performance of PARS. We design a training procedure where we discard the filtered noisy labels and ignore the negative learning losses on both the original and pseudo labels (*i.e.*"PARS w/o $\mathcal{L}_{NL}$ in either $\mathcal{L}_{raw}$ or $\mathcal{L}_{pseudo}$"). Overall, there is a consistent reduction in model performance across all noise types and levels as compared to the full model. To the best of our knowledge, this is the first evidence in literature that the filtered noisy labels by a sample selection strategy can be effectively included (rather than discarded) in order to consistently boost the model performance in LNL.

## 5 CONCLUSION

In this paper, we proposed a novel method for training with noisy labels called **PARS: P**seudo-Label **A**ware **R**obust **S**ample Selection, which takes advantage of self-training with pseudo-labels and data augmentation, and exploits *original and pseudo labels* with label-dependent noise-aware loss functions with help of robust sample selection. We conduct several experiments on multiple benchmark datasets to empirically evaluate the strengths of our proposed method in learning with noisy labels. In particular, PARS gains significant improvements over the state-of-the-art methods in challenging situations such as when high levels of noise are present. Furthermore, we also evaluate our method under a novel low-resource/semi-supervised learning with noisy labels setting, where only a subset of noisy labels are available during training, and PARS outperforms the baseline models by a significant margin. We believe this is a ubiquitous setting in the real world when it may be difficult to obtain large-scale weak labels potentially due to data privacy concerns. In the future, we are interested in exploring further this semi-supervised learning setting with noisy labels, and extending our model to other domains such as audio and NLP.

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

# A  APPENDIX

## A.1  NOISE ROBUST LOSS FUNCTIONS

In this section, we discuss the choice of two noise robust loss functions that are used in our experiments.

Wang et al. (2019b) and Ma et al. (2020) design variants of Cross-Entropy (CE) loss that are noise-tolerant. Wang et al. (2019b) propose the Symmetric Cross Entropy (SCE) Loss where they combine the Reverse Cross Entropy (RCE) loss with the CE loss, as -

$$\mathcal{L}_{SCE} = \underbrace{-\alpha.\sum_{k=1}^{K}[y]_k\log[p(\boldsymbol{x};\theta)]_k}_{\mathcal{L}_{CE}} \underbrace{-\beta.\sum_{k=1}^{K}[p(\boldsymbol{x};\theta)]_k\log([y]_k)}_{\mathcal{L}_{RCE}} \tag{12}$$

where $\alpha$ and $\beta$ are hyperparameters to control the effect of the two loss functions and $K$ are the total number of classes. Equation (12) is similar to the symmetric KL-divergence and as $y$ does not represent the true label distribution (due to noisy labels) the RCE loss term minimizes the entropy between the true and predicted distribution, $p(\boldsymbol{x};\theta)$, as this can reflect the true distribution to some extent. As argued by Ma et al. (2020), SCE loss is only partially robust to noisy labels and they introduce a normalized version of traditional loss functions for classification, *e.g.* Cross-Entropy (CE), Mean Absolute Error (MAE), Focal Loss (FL) and Reverse Cross Entropy (RCE).

For instance, the general normalized form for the Cross Entropy loss named as Normalized Cross Entropy (NCE) is formulated as -

$$\mathcal{L}_{NCE} = \frac{\mathcal{L}_{CE}(p(\boldsymbol{x};\theta),y)}{\sum_{k=1}^{K}\mathcal{L}_{CE}(p(\boldsymbol{x};\theta),k)} \tag{13}$$

Furthermore, they empirically show that although robust, these normalized loss functions suffer from underfitting. To deal with this, they use a combination of "Active" and "Passive" losses. Active Losses (CE, Normalized CE, FL and Normalized FL) maximize the network's output probability at the class position only specified by the label, whereas Passive losses (MAE, Normalized MAE, RCE, Normalized RCE) minimize the probability at atleast one other class position. Combining the passive loss with the normalized active loss such as normalized cross entropy loss helps to deal with the denominator increasing significantly in eq. (13). Hence, Active Passive Loss (APL) can leverage both the robustness and convergence advantages. Formally,

$$\mathcal{L}_{APL} = \alpha.\mathcal{L}_{Active} + \beta.\mathcal{L}_{Passive} \tag{14}$$

Four possible combinations satisfying the APL principle are considered in the original work - $\alpha\mathcal{L}_{NCE} + \beta\mathcal{L}_{MAE}$, $\alpha\mathcal{L}_{NCE} + \beta\mathcal{L}_{RCE}$, $\alpha\mathcal{L}_{NFL} + \beta\mathcal{L}_{MAE}$ and $\alpha\mathcal{L}_{NFL} + \beta\mathcal{L}_{RCE}$.

## A.2  ALGORITHM OF PARS

The detailed algorithm of PARS, **P**seudo-Label **A**ware **R**obust **S**ample **S**election, is summarized in Algorithm 1.

---

**Algorithm 1** PARS: **P**seudo-Label **A**ware **R**obust **S**ample **S**election

---

1: **Input:** Noisy labeled dataset $\mathcal{D} = \{(\boldsymbol{x}, y)^{(i)}\}_{i=1}^{n}$, confidence threshold $\tau$, self-training loss weight $\lambda_S$, negative learning loss weight $\lambda_N$, confidence penalty weight $\lambda_R$, current epoch $e$, total epochs $E$, warm-up epoch $w_e$ ($w_e < E$), original model parameters $\theta$ and model parameters after warmup $\hat{\theta}$.

2: **while** $e < w_e$ **do**

3:     $\min_\theta \frac{1}{|\mathcal{D}|} \sum_{(\boldsymbol{x},y)\in\mathcal{D}} \mathcal{L}_{RL}(p(\boldsymbol{x};\theta), y)$               ▷ Robust Warm-up training as in Equation (5)

4: **end while**

5: **for** $e = w_e$ to $E$ **do**

6:     // Lines 7-8 , Robust Sample Selection

7:     $\mathcal{D}_A = \left\{(\boldsymbol{x}, y) \in \mathcal{D} \middle| \max_k \left[p(\boldsymbol{x};\hat{\theta})\right]_k > \tau \right\}$               ▷ Ambiguous Label Set

8:     $\mathcal{D}_N = \mathcal{D} \setminus \mathcal{D}_A$               ▷ Noisy Label Set

9:     // Line 10 , Noise-Aware training with filtered raw labels

10:     $\mathcal{L}_{raw}(\theta) := \left(\frac{1}{|\mathcal{D}_A|} \sum_{(\boldsymbol{x},y)\in\mathcal{D}_A} \mathcal{L}_{RL}(p(\boldsymbol{x};\theta), y)\right) + \lambda_N \left(\frac{1}{|\mathcal{D}_N|} \sum_{(\boldsymbol{x},\bar{y})\in\mathcal{D}_N} \mathcal{L}_{NL}(p(\boldsymbol{x};\theta), \bar{y})\right)$

11:     // Lines 12-13 , Noise-Aware self-training with filtered pseudo-labels

12:     Predict positive pseudo-label, $z$ and negative pseudo-label. $\bar{z}$ as given in Equation (8)

13:     $\mathcal{L}_{pseudo}(\theta) := \left(\frac{1}{|\mathcal{D}_A|} \sum_{(\boldsymbol{x},\cdot)\in\mathcal{D}_A} \mathcal{L}_{PL}(p(\boldsymbol{x};\theta), z)\right) + \lambda_N \left(\frac{1}{|\mathcal{D}_N|} \sum_{(\boldsymbol{x},\cdot)\in\mathcal{D}_N} \mathcal{L}_{NL}(p(\boldsymbol{x};\theta), \bar{z})\right)$

14:     Calculate Confidence Penalty $\mathcal{L}_{reg}(\theta)$ given in Equation (10)

15:     $\mathcal{L}_{total}(\theta) := \mathcal{L}_{raw}(\theta) + \lambda_S \mathcal{L}_{pseudo}(\theta) + \lambda_R \mathcal{L}_{reg}(\theta)$               ▷ Final Loss

16: **end for**

17: **return** $\mathcal{L}_{total}(\theta)$

---

### A.3 IMPLEMENTATION DETAILS

**CIFAR-10/100.** We use PreAct Resnet-18 (He et al., 2016b) as our backbone model following previous works (Li et al., 2020). All networks are trained using SGD with momentum 0.9, weight decay 5e-4 and batch size of 128. The models are trained for 100 epochs. For the learning rate schedule, we use a cosine learning rate decay (Loshchilov & Hutter, 2016) which changes the learning rate to $\eta cos(\frac{7\pi t}{16T})$ where $\eta$ is the initial learning rate, $t$ is the current training step and $T$ are the total number of training steps. The initial learning is set to 0.03, $\lambda_S$ to 1.0, $\lambda_N$ to 0.1, $\lambda_R$ to 1.0, and the confidence threshold $\tau$ for sample selection is set to 0.95 unless stated otherwise. For the robust warm-up training, we train with the combination of Normalized Cross Entropy ($\mathcal{L}_{NCE}$) and Mean Absolute Error ($\mathcal{L}_{MAE}$). The weights for the NCE and MAE are 1.0 for CIFAR-10, and 10.0 and 1.0 respectively for CIFAR-100. The warm-up for CIFAR-10 is done for 25 epochs if the noise ratio is below 0.6, and 40 epochs if the noise ratio is above 0.6. For CIFAR-100, the warm-up is done for 50 epochs. For all experiments on CIFAR-10 and CIFAR-100 datasets, we sample data with three times the original batch size for the self-training loss. The confidence based thresholding ($\tau$ - 0.95) is also performed on this pseudo-labeled batch of data samples for the positive and negative learning. We apply Exponential Moving Average to the model parameters with decay of 0.999. We leverage RandAugment (Cubuk et al., 2020) for strong augmentations on the data samples, and keep the number of augmentations fixed to 2 for fair comparison. All our models are trained using the PyTorch framework (Paszke et al., 2019). The code will be made publicly available for future research purposes.

**Clothing1M.** The backbone network used for the Clothing1M dataset is ResNet-50 initialized with pre-trained ImageNet weights following previous work (Li et al., 2020). We train the network using SGD with momentum of 0.9, weight decay of 0.001 and a batch size of 32. The warm-up is done for 10 epochs with the Symmetric Cross Entropy loss ($\mathcal{L}_{SCE}$) with the same hyper-parameter settings as proposed in the paper (Wang et al., 2019b). The initial learning rate is set to 0.002, and decayed by a factor of 10 every 25 epochs. We set $\lambda_S$ to 1.0 and $\lambda_N$ to 0.01, $\lambda_R$ to 1.0. For strong augmentations, we use RandAugment (Cubuk et al., 2020) as in CIFAR datasets with the number of augmentations fixed to 4.

**Choice of Robust Loss** $\mathcal{L}_{RL}$**.** In each experiment, we test several different robust loss functions (Equation (2)) and then pick the best performing $\mathcal{L}_{RL}$ based on the validation loss for PARS. For fair comparison, we design a baseline that is trained using the selected robust loss function alone in each experiment, which is equivalent to performing the warm-up training thoroughly without the

rest of PARS (Equation (5)). Comparison of this baseline and the final PARS can demonstrate the importance of different components of our proposed method, PARS, independently of the choice of a robust loss function. Note that studying which noise robust loss is best suited for each dataset and/or noise setting is beyond the scope of this paper, which we leave for future work.

### A.4 ADDITIONAL EXPERIMENTS

**Convergence of Test Accuracy.** In Figure 3, we compare the test accuracies for our final model, *PARS* trained on all the original and pseudo labels vs. the model ablation without any negative learning losses on both the original and pseudo labels (*PARS w/o negative learning*). PARS outperforms the model trained without negative learning and achieves stable and better performance. This demonstrates the effectiveness of training with negative learning losses.

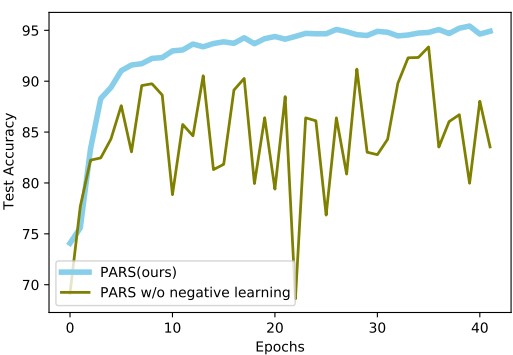

Figure 3: Test accuracy curves after robust warmup training on CIFAR-10 dataset with 40% asymmetric noise.

**Noise Robust Loss for Corrected Pseudo Labels.** As discussed in the main paper, the corrected pseudo labels are trained with a positive learning loss unlike the ambiguous original labels (which are trained with a noise robust loss). Here, we empirically evaluate the performance of the model on the CIFAR-10 dataset with 50% symmetric label noise when trained with noise robust loss for corrected pseudo labels. This model achieves the test accuracy of 92.9%, much lower than the accuracy achieved of 95.8% with the proposed PARS model. This shows that the model underfits when trained with the noise robust losses with clean/corrected pseudo labels. This is in line with what is observed in previous work (Ma et al. (2020)).

**Ablation study on Confidence Threshold, $\tau$.** Figure 4 shows the test accuracy for our final model PARS with 90% symmetric noise on the CIFAR-10 dataset at different confidence threshold for sample selection. The model accuracy drops below threshold 0.95 due to the poor quality of pseudo-labels (and original labels) in the separated corrected (or ambiguous) label sets. Setting a higher threshold keeps only the high-confident and reliable predictions and hence leads to a better performance.

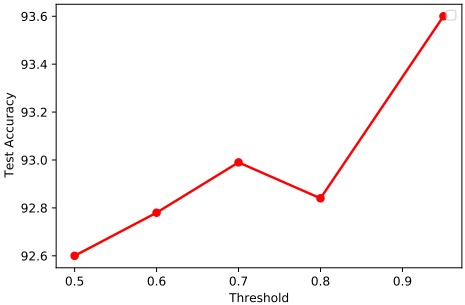

Figure 4: Test accuracy of PARS at different threshold rates on CIFAR-10 dataset with 90% symmetric noise.

