# OpenReview forum: "PARS: PSEUDO-LABEL AWARE ROBUST SAMPLE SELECTION FOR LEARNING WITH NOISY LABELS"
_ICLR.cc/2022/Conference — ICLR 2022 Submitted_

### Official Review · Reviewer_yGpG · 2021-11-02

**Correctness:** 4
**Technical Novelty And Significance:** 3
**Empirical Novelty And Significance:** 3
**Recommendation:** 6
**Confidence:** 4

**Main Review:**

__Strengths__
-	The paper address problems arising from the three main approaches used to tackle noisy labels. They discuss about the shortcomings of sample selection algorithms, loss functions designed for noisy distribution and label correction/pseudo-labelling algorithms. While all of them reduce noisy labels in successive iterations, a common problem stems from confirmation bias of the network due to higher noise in labels. The authors study and explain various ways to address this issue and propose a noise aware loss mechanism that adjust the weight for ambiguous and noisy labels, reducing the training error from prior approaches.
-	For sample selection mechanism, the authors use the confidence of all classes for thresholding and decide based on the highest probability value of all present classes. This is different from prior works which use the probability value of raw label (which could be noisy pseudo-label) class for thresholding and deciding if the sample is noisy or clean. The authors’ proposed mechanism only uses the network’s output for separating the samples into ambiguous and noisy set and can reduce noise propagation from one iteration to the other.
-	The use of different loss functions for different sample set based on the noise level is also intuitive to adjust the impact they have during the training process. Dealing with ambiguous and noisy labels with independent losses allows adjustment of feedback from these labels better. This is one of the valid methods to handle learning from noisy labels which this paper sheds more light on and their results in CIFAR-10/100 validates the claim. Clothing1M scores are similar to prior work which hints that this loss formulation is equally valid as previous methods.
-	The proposed method shows better resilience to noisy labels when learning in a low-resource semi-supervised setting and significantly outperforms prior works. Compared to FixMatch which is dedicated to learn using semi-supervised approach on low resource, the proposed approach is able to handle unlabelled data better and reduce impact from noise.

__Weakness__
-	The authors specify they use pseudo-labels from both D_A and D_N in Eq 9 to train the network. Is this pseudo-label based self-training done together with noise aware training with raw labels? Or is it trained in alternating iterations with either of those losses?
-	Continuing on previous point, a large gain is seen with the use of pseudo-labels in training which suggests the model’s prediction after warm-up is better than the raw labels. In that case, the raw labels might differ from the pseudo-labels significantly. The small performance gap by removing negative learning in both pseudo and raw loss shows its more dependent on the positive learning using perceived labels. In this case, the network is being trained using both raw label and pseudo-label for the same sample at the same time, which could be different from each other. Is there a detailed explanation for this occurrence?
-	Have the authors tried not using raw labels after warm-up and only use the pseudo-labels for all loss formulations? This is interesting as biggest gain comes from pseudo-labels after the initial warm-up.
-	From appendix A.4, using noise robust loss on pseudo-labels performs lower than using positive learning from Eq 9. This finding is also interesting as the pseudo-label would also be assumed to have certain ambiguity to it that could benefit from the noise robust loss. Is the model underfit caused by pseudo-labels being very accurate that positive learning has better feedback than a toned-down robust loss?
-	Since the approach shows more impact in semi-supervised setting, perhaps including more study on that similar to FixMatch would benefit the paper and the readers more.


**Summary Of The Paper:**

This paper proposes a unified approach to handle noisy labels for training neural networks and utilize all the training data to learn effectively when noise is present. The authors utilize the assumed correct and noisy labels with different loss functions adjusted using different weights in order to adjust the impact on training. The authors show how their approach improves over CIFAR-10 and CIFAR-100 with high noise ratio and demonstrate competitive results in Clothing1M dataset. The authors also show strong results in semi-supervised setting, outperforming FixMatch significantly.

**Summary Of The Review:**

The paper provides a unified algorithm to learn from noisy label and shows convincing results on various datasets. The results on semi-supervised setting are also appreciated and can be useful for future research in this direction.

---

> ### Author Response · Authors · 2021-11-18
> **Response to Reviewer yGpG**
>
> We thank the reviewer for their thoughtful comments, feedback and acknowledging our contributions!  Below are the pointwise responses to clarify the comments raised. We will be happy to  engage in a discussion if there are still some concerns.
>
> > The authors specify they use pseudo-labels from both D_A and D_N in Eq 9 to train the network. Is this pseudo-label based self-training done together with noise aware training with raw labels? Or is it trained in alternating iterations with either of those losses?
>
> The model is trained together with both the raw and pseudo-labels as in Eq. 11. We have also made this clear in the caption of Figure 1 (modifications in red).
>
>
> > Training on raw labels and pseudo-labels together might lead to training on two different labels at the same time, what happens then?
>
> This is a great observation! The pseudo-labels can be cleaner than the raw labels, hence there is a chance that positive learning happens on two different sets of labels. Thanks to robust learning on the raw labels for the “expected-to-be-clean” set, it prevents overfitting to the noisy labels in case of divergent labels.
>
> Moreover, the model receives a stronger training signal from the positive learning loss on pseudo-labels as compared to the training on the raw noisy labels (due to the robust learning loss).
>
>
> > Experiments with only training on pseudo labels and no raw labels.
>
> Thanks for suggesting this ablation! We run some additional experiments to show the performance when training without raw labels. The performance gain with PARS in 50%/90% sym. noise over training without raw labels is indicative of the importance of joint training with raw and pseudo labels. Please see results below.
>
> | CIFAR-10      | 50% Sym. Noise | 90% Sym. Noise |
> | ----------- | ----------- |  ----------- |
> | Robust Warmup      | 76.4       | 54.5 |
> | PARS w/o L_raw   | 94.0 | 79.7 |
> | PARS  | 95.8 | 93.6 |
>
> > Pseudo labels undefit with noise robust loss, is it because they are more accurate?
>
> This is an interesting observation and also supports previous work in noise robust loss functions. For instance, Ma et al. [1] experimentally show that the model performance drops with noise robust loss when trained on a dataset with no noise.
> Under high-noise setting, pseudo-labels can be cleaner than the raw labels, under low-noise setting, the reverse may be true; overall this is one the main motivations that we work on using both raw and pseudo labels on all data, so our model is capable of handling a wide range of noise level.
>
> > More analysis on the semi-supervised setting.
>
> An in-depth analysis and further exploration of LNL in low resource is an interesting direction but not the main focus of this work. We hope our work will open new research avenues in this domain.
>
> [1] Ma, Xingjun, et al. "Normalized loss functions for deep learning with noisy labels." ICML, 2020.

---

### Official Review · Reviewer_mAAg · 2021-11-02

**Correctness:** 3
**Technical Novelty And Significance:** 2
**Empirical Novelty And Significance:** 2
**Recommendation:** 5
**Confidence:** 3

**Main Review:**

Strengths:
1. This paper proposes an easy-to-implement method PARS for LNL task and it obtains certain improvements under the traditional LNL setting especially with a large noise ratio and a low-resource semi-supervised LNL setting. The ablation study also shows the contributions of each of the two main improvements in this paper.
2. The paper is well organized and easy to follow, with clear explanations of the algorithm details.

Weaknesses:
1. This paper mainly makes use of two existing techniques, i.e., negative learning and FixMatch (a semi-supervised learning method) to build a better framework. While the motivation of adopting negative learning is relatively clear, I think the authors should further explain the motivation of applying the pseudo-labeling for the entire datasets instead of only the noisy data, the reason of using strong data augmentation in this step, and the difference compared to FixMatch. Since the pseudo-labeling contributes a lot to the results as shown in Table 4, further discussions and analyses are needed to figure out why the pseudo-labeling is so important in PARS? Are the pseudo labels more accurate than original labels or does the consistency between the model predictions of a sample with weak and strong augmentations matter? I think some in-depth analyses lack here. Besides, the authors could try replacing the strong augmentation with weak augmentation to show its advantage with experimental results.
2. In my opinion, when the noise ratio becomes large, from the perspective of PARS, the LNL task becomes more similar to a semi-supervised learning task, thus PARS (part of which inherits from FixMatch) would achieve a high performance as shown in Table 1. However, when the noise ratio is small, i.e., 20% or 50%, PARS doesn’t perform the best on CIFAR-100 dataset. I think the reason of this situation should be further discussed. Also, on Clothing1M dataset, is the fact that PARS doesn’t perform the best possibly related to the noise ratio in this dataset?
3. Some claims or components of PARS are not evaluated. (1) In Equation 6, a label-free conﬁdence-based thresholding is proposed and the authors claim that it would reduce the self-conﬁrmation bias. Are there any empirical results that would support this claim? (2) I think an additional ablation study is needed for validating the effectiveness of Equation 10.


**Summary Of The Paper:**

This paper proposes a hybrid framework PARS for Learning with Noisy Labels (LNL) task. The framework jointly leverages the original noisy labels and estimated pseudo labels of all samples for model training. Specifically, for samples whose maximum classification probabilities are higher than a threshold, their original/pseudo labels are used in robust/positive learning, while for the remaining samples, their original/pseudo labels are used in negative learning. When using pseudo labels, strong augmentations are also applied to the samples. Experiments conducted on three public datasets with the traditional and a new low-resource semi-supervised LNL settings show certain improvements over existing methods.

**Summary Of The Review:**

This paper unifies several well-performed methods for LNL task to build a better framework, which achieves certain improvements especially under high-noise scenarios. However, some in-depth analyses about the effectiveness of the pseudo-labeling in PARS and the inferior performances under some settings still lack in this paper.

---

> ### Author Response · Authors · 2021-11-18
> **Response to Reviewer mAAG**
>
> We thank the reviewer for their thoughtful comments, feedback and acknowledging our contributions!  We hope our pointwise responses below will address the concerns. We will be happy to  engage in a discussion if there are still some concerns.
>
> ### Why pseudo-labeling for entire data instead of just noisy ones?
>
> The key motivation for using pseudo-labels on all data stems from the observation that it is rather complex to divide the samples into perfectly clean and noisy sets. For instance, DivideMix separates the two sets by unsupervised loss modeling that introduces significant overhead.
> We seek to avoid this by relaxing the constraint on sample selection using label-dependent noise aware loss functions and self-training with pseudo-labels. Pseudo-labeling on only the noisy set discards the samples that might still be noisy in the “expected to be clean set”. Our approach exploits all the data so that our model is capable of handling a wide range of noise level.
>
> ### Differences to FixMatch.
>
> Indeed, PARS is reminiscent of FixMatch, but our goal is improving robustness to label noise with the following contributions and differences.
> 1) FixMatch is only trained with a positive loss above a threshold, but PARS adds the negative learning loss component for the samples below the threshold. This is significantly important when the samples learned from are noisy. Fixmatch alone doesn’t perform well with label noise (Table 3) which advocates our proposed noise-aware label-dependent treatments to both the raw and the pseudo-labels.
>
> 2) We don’t exploit SSL in a traditional way as in FixMatch or DivideMix. DivideMix trains with pseudo-labels on the noisy set, instead we obtain pseudo-labels for the entire dataset without relying on separation of clean and noisy (unlabeled/pseudo-labeled) and show significant improvements.
>
> 3) As FixMatch operates in a low resource setting, it assumes a large batch of unlabeled data to smooth the decision boundary by enforcing consistency between different views of the same input (augmented samples). The primary motivation behind augmentations in PARS for LNL is to prevent overfitting to wrong pseudo-labels and improve generalization without the need for a large unlabeled batch.
>
> ### Advantages of strong data augmentation. Empirical results by replacing strong with weak augmentation.
>
> In order to prevent overfitting to wrong pseudo-labels and accumulation of errors in each training iteration, we use strong data augmentations as a regularizer. This is also inspired from previous work [4] and has shown to be really effective to resolve self-confirmation bias while pseudo-labeling.
>
> Empirically, the accuracy drops down to 91.7% from 95.8% for CIFAR-10 dataset with 50% symmetric noise (Table 1) when training with weak augmentation. Playing with different augmentation strategies is a great suggestion and we plan to explore this for future work!
>
> >  Are the pseudo labels more accurate than original labels or does the consistency between the model predictions of a sample with weak and strong augmentations matter?In depth analysis of pseudo-labels?
>
> The pseudo labels can be more accurate than original labels (especially in high-noise settings). Replacing strong augmentation with weak (empirical results given above), the model performance does drop slightly but we don’t see a big divergence in training.
> Whereas, the model training diverges in semi-supervised learning when weak aug. is replaced with strong aug. defeating the purpose of consistency regularization.
> In our case to further improve the quality of pseudo-labels and performance, we observe that strong augmentations perform better.
>
> > when the noise ratio becomes large, the LNL task becomes more similar to a semi-supervised learning task.
>
> We beg to differ here as the gains in PARS don’t only come from the pseudo-labels but also from training on raw labels.
> For empirical support, we provide results in our response to reviewer yGpG.
> If we only train with pseudo-labels (PARS w/o L_raw), then the performance drops in high-noise ratios.
>
> ### PARS performance with small noise ratio and clothing1m is small.
>
> Thanks for bringing this up for discussion! Please see our responses to reviewer NCaX for a detailed response.
>
>
> [1] Li, Junnan, et al. "DivideMix: Learning with Noisy Labels as Semi-supervised Learning." ICLR, 2019.
> [2] Arazo, Eric, et al. "Unsupervised label noise modeling and loss correction." ICML, PMLR, 2019.
> [3] Tanaka, Daiki, et al. "Joint optimization framework for learning with noisy labels." CVPR, 2018.
> [4] Arazo, Eric, et al. "Pseudo-labeling and confirmation bias in deep semi-supervised learning." IJCNN, IEEE, 2020.

---

> > ### Author Response · Authors · 2021-11-18
> > **Response to Reviewer mAAG**
> >
> > ### Ablation study without label-free conﬁdence-based thresholding to support the claim for reducing self-conﬁrmation bias.
> >
> > Thank you for suggesting this ablation! We run additional experiments and report results with label-aware thresholding for the original/raw label sample selection.
> > The accuracy drops to 72.3% (best accuracy)  from 76.4% (after warm-up) on the CIFAR-10 dataset with 50% sym. noise. The model tries to achieve a lower loss by filtering all the labels in the noisy set (below the threshold). This harms further learning from the robust and self-training loss.
> >
> > ### Ablation study for Eq.10.
> >
> > This is a common technique used in [1,2], which was first designed to prevent the generated pseudo-labels from collapsing to a trivial local minima (Sec 4.2 [3]) which would have a fatally negative impact on model convergence.
> > Without the regularization, the model performance drops to 26% (CIFAR-10 with 90% sym. noise) from 54% (after warm-up) in a few training epochs.
> >
> > [1] Li, Junnan, et al. "DivideMix: Learning with Noisy Labels as Semi-supervised Learning." ICLR, 2019. [2] Arazo, Eric, et al. "Unsupervised label noise modeling and loss correction." ICML, PMLR, 2019. [3] Tanaka, Daiki, et al. "Joint optimization framework for learning with noisy labels." CVPR, 2018. [4] Arazo, Eric, et al. "Pseudo-labeling and confirmation bias in deep semi-supervised learning." IJCNN, IEEE, 2020.

---

### Official Review · Reviewer_NCaX · 2021-11-03

**Correctness:** 3
**Technical Novelty And Significance:** 3
**Empirical Novelty And Significance:** 3
**Recommendation:** 6
**Confidence:** 4

**Details Of Ethics Concerns:**

No concern

**Main Review:**

Strengths
- S1: A novel combination of the existing successful approaches with a combination of novel contributions in some components (e.g., sample selection)
- S2: Good empirical gains over LNL SOTA methods (Table 3)

Weaknesses
- W1: Fig 1 is not clear. As the method seems having the order of operations, the figure without the order of operations is not clear. In addition, it is not clear how the pseudo-label generator is used to label the noisy labeled data.
- W2: Empirical gain in small noise regime (SYM 20%-50% in CIFAR-10/100) are small.
- W3: No empirical gain against SOTA in large dataset (Clothing1M)
- W4: There are large gains in large noise regime (more than 80%). But this set-up may not be very realistic.

**Summary Of The Paper:**

The paper combines three branches of approaches (1. sample selection, 2. noise robust loss, 3. label correction) to address label noise in classification in a single framework. Specifically, the method includes (1) warm up phase, (2) a novel label-free sample selection, (3) noise aware loss (as a standard technique) and (4) self-training with pseudo labels along with the given labels. The proposed method outperforms prior arts in CIFAR-10/100, especially in high noise regimes (80-90%). More interestingly, in the small sample training regime, gains by the proposed method increase significantly in evaluations with CIFAR-10/100.

**Summary Of The Review:**

The proposed method is a novel combination of existing successful components for LNL. Although the proposed method is somewhat novel, the empirical gain does not assert the benefit of the proposed approach. In addition, some of the presentation (e.g., Fig 1) is not clear to understand the proposed method as it omits the order of the procedure.

---

> ### Author Response · Authors · 2021-11-18
> **Response to Reviewer NCaX**
>
> We thank the reviewer for their thoughtful comments, feedback and acknowledging our contributions! We hope our pointwise responses below will address the concerns. We will be happy to engage in a discussion if there are still some concerns.
>
> ### Clarity in Fig 1.
> > W1: Fig 1 is not clear. As the method seems having the order of operations, the figure without the order of operations is not clear.
>
> Thanks for pointing this out for better clarity. We have updated the caption of Fig1. (modifications in red) to resolve the ambiguity of order of training in our method.
>
> > In addition, it is not clear how the pseudo-label generator is used to label the noisy labeled data.
>
> After the robust warmup training, we iteratively obtain the pseudo-labels on-the-fly for the noisy labeled data i.e. using the model at each iteration progressively during training. To be specific, the first set of pseudo labels are obtained via the updated model parameters from warmup, then these pseudo-labels are updated as training continues on both the raw and pseudo labels.
>
> > W2: Empirical gain in small noise regime (SYM 20%-50% in CIFAR-10/100) are small.
>
> Thank you for bringing this up for discussion!
>
> In the low noise regime, as also observed by Nishi etal [1] the model that performs better in high noise, does not necessarily perform well in low-noise settings. Their performance gains in the low noise regime in CIFAR-10/CIFAR-100 come from a strongly augmented warmup on top of the contributions from DivideMix, whereas PARS is a standalone and effective method in itself. Moreover, our focus is to study and provide a new tool and technique to tackle noisy labels rather than designing an over-engineered solution with carefully designed augmentation and hyperparameters. We leave this to future work to be used in conjunction with PARS.
>
>
> > W3: No empirical gain against SOTA in large dataset (Clothing1M)
>
> Similar to the low noise regime, Clothing1M dataset as stated by Nishi etal [1] has a relatively low noise ratio because the low-noise variant of their method achieves the highest performance on this dataset. In particular, most SOTA methods on this dataset atmost gain <0.5-1% improvement. While training on standard CE achieves around 69% accuracy, the best method so far (DivideMix-based DM-AugDesc-WS-SAW) with semi-supervised learning, carefully designed augmentations and hyperparameter optimization and confidence penalties, achieves around 75%.
> Clothing1M being one of the benchmark datasets makes it easy to fairly compare with others. In the future we plan to experiment with more datasets with relatively high levels of label noise and in domains other than vision.
>
> ### Why is high-noise ratio important?
>
> > W4: There are large gains in large noise regime (more than 80%). But this set-up may not be very realistic.
>
> This is an interesting discussion for the LNL community! Thanks for bringing it up.  High noise ratio has been studied extensively in previous work [2,3,4] to study the effectiveness of noise robust models. This setting is particularly important to develop robust machine learning methods that can be deployed in contexts where it is rather hard to gather annotations [5].
> Our motivation particularly stems from rule-based weak label generation [6] where the noise ratio can be extremely high in the training dataset.
>
>
> [1] Nishi, Kento, et al. "Augmentation strategies for learning with noisy labels." CVPR, 2021.
>
> [2] Tanaka, Daiki, et al. "Joint optimization framework for learning with noisy labels." CVPR, 2018.
>
> [3] Li, Junnan, Richard Socher, and Steven CH Hoi. "DivideMix: Learning with Noisy Labels as Semi-supervised Learning." ICLR, 2019.
>
> [4] Arazo, Eric, et al. "Unsupervised label noise modeling and loss correction." ICML, PMLR, 2019.
>
> [5] Ghassemi, Marzyeh, et al. "A review of challenges and opportunities in machine learning for health." AMIA Summits on Translational Science Proceedings 2020.
>
> [6] Ratner, Alexander, et al. "Snorkel: Rapid training data creation with weak supervision." International Conference on Very Large Data Bases, 2017

---

> > ### Comment · Reviewer_NCaX · 2021-11-30
> > **Thank you for clarifying the concerns.**
> >
> > The reviewer appreciates the authors' responses to clarify most of my concerns. I have one last point for improvement: Fig.1: I intended to ask the authors to revise the figure (*e.g.*, putting number label in each component in Fig.1) instead of revising the caption.
> >
> > Given the novelty of the combination and less novelty of used components and some weaknesses in empirical results, the reviewer believe the paper is still marginally in the acceptance side.

---

> > > ### Author Response · Authors · 2021-12-02
> > > **Response to response**
> > >
> > > We thank the reviewer for providing additional feedback, which allows us to understand what the major concerns remain. We will revise Fig 1 as suggested and again thank the reviewer for identifying our novelty and contributions.

---

### Decision · Program_Chairs · 2022-01-20

**Decision:**

Reject

**Comment:**

This paper a framework of learning with noisy labels named PARS that combines three types of approaches, i.e., sample selection, noise robust loss, and label correction.  The framework leverages both original noisy labels and estimated pseudo labels of all samples for improving the training performance, and the empirical studies demonstrated competitive results on CIFAR datasets especially in high-noise and low-resource settings.

Reviewers raised some major concerns about the weaknesses. For example, empirical gain in small noise regime are small or negligible, and no empirical gain against SOTA in large dataset with real-world noise (Clothing1M).  While large gains in large noise regime (more than 80%), such setting may not be very realistic and there also  lack of in-depth analysis on the sources of the gain (e.g., it is unknown if the gain is mainly because of using a better SSL or other factors since LNL becomes more similar to SSL when noise is very high). For technical novelty perspective, while the proposed approach is new, the overall novelty may not be very significant as this paper mainly combines existing techniques, e.g., negative learning and FixMatch (a semi-supervised learning method) in the proposed learning approach.

Authors have made great efforts for addressing the reviewers’ concerns partly, but some major concerns on the technical novelty and empirical studies remain.  Therefore, the paper is not recommended for acceptance in its current form. I hope authors found the review comments and discussions useful and constructive, and like to see it accepted in the near future after these issues are fully addressed.